# Short- vs. Long-Course Antibiotic Treatment for Acute Streptococcal Pharyngitis: Systematic Review and Meta-Analysis of Randomized Controlled Trials

**DOI:** 10.3390/antibiotics9110733

**Published:** 2020-10-26

**Authors:** Anna Engell Holm, Carl Llor, Lars Bjerrum, Gloria Cordoba

**Affiliations:** Research Unit for General Practice and Section of General Practice, Department of Public Health, Øster Farimagsgade 5, 1014 Copenhagen, Denmark; carles.llor@gmail.com (C.L.); lbjerrum@sund.ku.dk (L.B.); gloriac@sund.ku.dk (G.C.)

**Keywords:** systematic review, meta-analysis, streptococcal pharyngitis, antibiotic treatment, antimicrobial resistance

## Abstract

BACKGROUND: To evaluate the effectiveness of short courses of antibiotic therapy for patients with acute streptococcal pharyngitis. METHODS: Randomized controlled trials comparing short-course antibiotic therapy (≤5 days) with long-course antibiotic therapy (≥7 days) for patients with streptococcal pharyngitis were included. Two primary outcomes: early clinical cure and early bacterial eradication. RESULTS: Fifty randomized clinical trials were included. Overall, short-course antibiotic treatment was as effective as long-course antibiotic treatment for early clinical cure (odds ratio (OR) 0.85; 95% confidence interval (CI) 0.79 to 1.15). Subgroup analysis showed that short-course penicillin was less effective for early clinical cure (OR 0.43; 95% CI, 0.23 to 0.82) and bacteriological eradication (OR 0.34; 95% CI, 0.19 to 0.61) in comparison to long-course penicillin. Short-course macrolides were equally effective, compared to long-course penicillin. Finally, short-course cephalosporin was more effective for early clinical cure (OR 1.48; 95% CI, 1.11 to 1.96) and early microbiological cure (OR 1.60; 95% CI, 1.13 to 2.27) in comparison to long-course penicillin. In total, 1211 (17.7%) participants assigned to short-course antibiotic therapy, and 893 (12.3%) cases assigned to long-course, developed adverse events (OR 1.35; 95% CI, 1.08 to 1.68). CONCLUSIONS: Macrolides and cephalosporins belong to the list of “Highest Priority Critically Important Antimicrobials”; hence, long-course penicillin V should remain as the first line antibiotic for the management of patients with streptococcal pharyngitis as far as the benefits of using these two types of antibiotics do not outweigh the harms of their unnecessary use.

## 1. Introduction

Acute pharyngitis is one of the most common complaints that a physician encounters in the ambulatory care setting, accounting for 1% to 2% of all ambulatory care visits annually and a high antibiotic prescribing rate [1,2]. However, the majority of these cases are viral and are self-limiting even in cases caused by bacteria belonging to group A streptococcus (GAS), namely “Streptococcus pyogenes” [3]. Nonetheless, there is consensus worldwide that antibiotic treatment is indicated for those with a high probability or confirmed GAS infection [4,5], specifically in high risk patients (pregnancy, HIV infection, immune deficiency).

The spread and development of antimicrobial resistance has called attention to the urgent need to optimize the use of antibiotics. Hence, the debate of whether short-term antibiotic therapy is better than long-term antibiotic treatment has gained momentum [6]. However, lack of evidence on the clinical effectiveness of short courses compared to long courses hampers the possibility to develop and apply novel recommendations in daily practice.

There is wide variation in clinical guidelines regarding management of acute pharyngitis caused by GAS [7]. Nonetheless, the use of a 10-day course (long course) with penicillin V is still widely accepted as a first line treatment option, as recommended by the current American and European guidelines [4,5]. Penicillin-resistant GAS has never been documented [8]. However, amoxicillin is often used in place of penicillin V as the first choice in some situations: 1. for young children as the efficacy appears to be equal, but this choice is primarily related to acceptance of the taste of the suspension; 2. shortage of penicillin in some countries; and 3. advantage of once-daily dosing [9,10], which may enhance adherence, and is relatively inexpensive.

There is conflicting opinions regarding the effectiveness and safety of shifting towards a shorter course. First of all, seven systematic reviews comparing short-course vs. long-course agree that short-course antibiotic treatment is non-inferior compared to long-course antibiotic treatment regarding clinical cure [11,12,13,14,15,16,17]. The majority of the studies included in the systematic reviews compare a short course of broad-spectrum antibiotics such as macrolides and cephalosporins with a long course of penicillin V (i.e., a narrow spectrum antibiotic). Focusing attention only on clinical effectiveness is problematic because the use of broad-spectrum antibiotics favors the development of antimicrobial resistance. Furthermore, a systematic review found that short antibiotic courses for streptococcal pharyngitis are inferior to eradicate GAS at end-of-therapy [18], in line with a recent trial published last year, which compared short vs. long courses of penicillin V in patients with GAS pharyngotonsillitis [19]. Based on this, it remains unclear whether short courses are a good option in comparison to long courses in the management of patients with pharyngitis caused by GAS.

This study aims to assess the clinical and bacteriological effectiveness and safety of short-term antibiotic therapy in comparison with long-term antibiotic regimens for the management of GAS pharyngitis in patients seeking care in primary care.

## 2. Methods

### 2.1. Types of Studies

Randomized controlled trials compared short-term vs. long-term antibiotic courses. We excluded studies in languages other than English, French, Spanish and German. Furthermore, we excluded studies comparing antibiotics with another type of treatment or patients requiring hospitalization at enrolment to treatment.

### 2.2. Types of Participants

All patients (i.e., adults and children) managed in primary care with confirmed GAS pharyngitis. The diagnosis of GAS had to be confirmed by a positive throat culture, rapid test (antigen detection test) or both.

### 2.3. Types of Interventions and Outcomes

Short-term (5 days or less) of antibiotic therapy and standard longer courses (7 days or more). Two primary outcomes were considered: early clinical cure, defined as the absence of fever and/or persistent pharyngeal pain within two weeks after completion of antibiotic treatment; early bacterial eradication, defined as a negative culture of a throat swab obtained within two weeks after completion of antibiotic treatment. The secondary outcomes were late clinical cure, defined as the absence of fever and/or persistent pharyngeal pain two weeks after completion of antibiotic treatment; late bacterial eradication, defined as a negative culture of a throat swab or recurrence obtained at least two weeks after completion of antibiotic treatment; and adverse events, such as diarrhea and rash.

### 2.4. Search Strategy and Selection Criteria

We searched in PUBMED/Medline (January 1966 to December 2019). We used a broad search string to increase the sensitivity of the search [20]. The following search terms were used: (((((“Anti-Bacterial Agents”[Mesh]) AND antibiotic*[Text Word])) AND ((((“Pharyngitis/drug therapy”[Mesh]) OR sore throat*[Text Word] OR pharyngitis[Text Word]) OR tonsillopharyngitis[Text Word]) OR throat, sore[Text Word])) AND (“Tonsillitis/drug therapy”[Mesh] OR sore throat*[Text Word] OR tonsillitis[Text Word]) OR tonsillopharyngitis[Text Word]) OR throat, sore[Text Word]. Additionally, we hand-searched reference lists of all the articles identified by the above-mentioned methods.

## 3. Data Collection

Literature search, screening of title and abstract were independently performed by two reviewers (AEH, CL). The program Covidence^®^ (Melbourne, Australia) was used for screening. Duplicates were removed, and articles were selected according to our inclusion criteria. After this step, full-text reports were assessed for eligibility. Any discrepancies were discussed with a third reviewer (GC). We extracted data on the number of participants, age, gender, diagnostic criteria and sponsorship. To assess differences in the intervention, we extracted data on type of antibiotic, dose, schedule and length. Some papers compared more than one short- or long-term course of antibiotics. In that case, we added up all the participants assigned to these courses for the main analyses. In other cases, different doses of the same short-term antibiotics were evaluated, and similarly, they all were included for the analyses. We also extracted data on microbiological results and reporting of adverse events.

### 3.1. Risk of Bias

The methodological quality of the studies was assessed according to risk of bias using the Cochrane Risk-Of-Bias (ROB) tool [21]. Risk of bias was labelled as: low (methods clearly described and adequate), high (methods described and inadequate) or unclear (insufficient information to assess the quality of the methods). Quality was assessed in the following domains: sequence generation, allocation concealment, blinding of participants/personnel, blinding of outcome assessors, incomplete outcome data and selective outcome reporting. To assess selective outcome reporting, we searched the World Health Organization International Clinical Trial Registry Platform (WHO ICTRP) [22] and the US National Institute of Health Ongoing Trials Register for completed and ongoing trials (clinicaltrials.gov) (Bethesda, MD, USA).

### 3.2. Analysis

The primary and secondary outcomes are dichotomous; hence, they are presented as the odds ratio (OR) with 95% confidence intervals (CIs). We analyzed only the available data and contacted authors to ask for further data collected in this century but did not contact authors of papers published more than 20 years ago. We tested for heterogeneity using the z score, chi^2^ test and I^2^ test statistics with values greater than 50% indicating substantial heterogeneity. Meta-analyses of the primary and secondary outcomes were performed using a random-effect model, since we expected a high degree of variability between the included studies. We used intention-to-treat (ITT) when data were available (i.e., the number of participants randomized was used as the denominator for each outcome). We performed subgroup analysis for trials with: (a) short-course penicillin vs. long-course penicillin, (b) short-course macrolides vs. long-course penicillin and (c) short-course cephalosporins vs. long-course penicillin. The statistical analyses were performed in the program Review Manager v5.3^®^ [23], which applies the Cochran–Mantel–Haenszel method for meta-analyses.

## 4. Results

The MEDLINE search yielded 1053 articles. A total of 968 studies were excluded based on title and 28 studies were also excluded based on abstract (Figure 1). Three more papers were excluded as they were subreports of papers already included in the review. Hence, 50 randomized clinical trials were included in the study.

### 4.1. Characteristics of the Included Studies

Table 1 shows the characteristics of the included studies. The oldest report is from 1972, while the latest report is from 2019 [24,25,26,27,28,29,30,31,32,33,34,35,36,37,38,39,40,41,42,43,44,45,46,47,48,49,50,51,52,53,54,55,56,57,58,59,60,61,62,63,64,65,66,67,68,69,70,71]. The included studies investigated a total of 19,004 patients. A total of 46 studies were published in English, three in French and one in Spanish. Five clinical trials compared short vs. long courses of penicillin, while the other 45 studies compared a total of 48 short courses of broad-spectrum antibiotic therapy: 28 with macrolides, 16 with cephalosporins, 3 with amoxicillin and clavulanate and one with a lincosamide. The most commonly antibiotic used for comparison was penicillin V, in 36 studies, of which 33 considered a 10-day course and the other three used a 7-day regimen. Twenty-four (48%) of the studies were financed by private companies, 22 (44%) did not report the funding sources and 4 (8%) were publicly funded. In the quality assessment, we found great weaknesses. Although all trials were randomized, up to 75% of the trials did not sufficiently describe the procedures to properly classify the risk of bias from the random sequence generation. The majority of the included studies had high risk of bias regarding blinding of participants and personnel, as well as blinding of outcome assessment (Figure 2).

### 4.2. Effects of Intervention

Primary outcomes: A total of 47 clinical trials reported data on early clinical cure, involving 18,581 patients. Overall, short-term antibiotic therapy was as effective as long antibiotic courses (OR 0.95; 95% CI, 0.79 to 1.15) (Figure 3). Three studies including 783 patients, which compared short-course penicillin vs. long-course penicillin, favored long-course penicillin for early clinical cure (OR 0.43; 95% CI, 0.23 to 0.82) (Appendix A). Short-term macrolide therapy was as effective as long-term penicillin courses in 17 studies including 5059 patients (OR 0.93; 95% CI, 0.68 to 1.26) (Appendix A). In 11 trials including 4282 patients, short-term cephalosporin therapy was associated with greater odds for clinical cure compared to long-term penicillin therapy (OR 1.48; 95% CI, 1.11 to 1.96) (Appendix A).

Early bacteriological eradication was available in 47 clinical trials, with a total of 17,659 individuals. The overall summary favored long-course therapy, although no statistical difference was observed (OR 0.78; 95% CI, 0.60 to 1.00). The Appendix A show the following subgroup analyses: (a) short course penicillin was associated with lower bacteriological eradication in comparison to long course penicillin (OR 0.34; 95% CI, 0.19 to 0.61); (b) short-term macrolide therapy was as effective as long-term penicillin regarding early bacteriological eradication (OR 0.76; 95% CI, 0.48 to 1.20); and (c) short-term cephalosporin regimens achieved greater eradication rates compared to long-term penicillin therapy (OR 1.60; 95% CI, 1.13 to 2.27).

Secondary outcomes: A total of 28 studies, including 11,853 patients reported data on late clinical cure. There was no difference between the two groups (OR 0.91; 95% CI, 0.80 to 1.04) (Appendix A). We observed no association among patients allocated to longer regimens (OR 0.81; 95% CI, 0.63 to 1.04) (Appendix A). Thirty-nine studies reported data on adverse events, including 14,081 patients. Adverse events were observed in 1211 (17.7%) patients assigned to short-term antibiotic regimens (17.7%), while 893 (12.3%) patients in the group assigned to long-term antibiotic regimens reported adverse events (OR 1.35; 95% CI, 1.08 to 1.68) (Appendix A). However, when different courses of penicillin were compared, more moderate adverse events were observed among patients taking the 10-day course in comparison to the five-day group—33% vs. 23%, respectively—information only reported in one trial [19].

## 5. Discussion

### 5.1. Summary of Main Results

A total of 50 randomized clinical trials were included in this article, constituting the most comprehensive and extensive systematic review of only randomized clinical trials published to date in patients with streptococcal pharyngitis. Most of the trials included in this review evaluated the effectiveness of short-course broad spectrum antibiotics (i.e., macrolides and cephalosporins), compared with a long course of the narrow spectrum antibiotic (penicillin V). Subgroup analysis demonstrated important differences regarding the effectiveness for early clinical cure and early microbiological cure depending on the antibiotic groups included in the comparison: (a) short-course penicillin was less effective when compared to long-course penicillin V; (b) short-course macrolides were equally effective when compared to long-course penicillin V; (c) short-course cephalosporins were more effective when compared to long-course penicillin V.

### 5.2. Strengths and Weaknesses of Study

The greatest strength of this review was our comprehensive search strategy. To our knowledge, this is the most comprehensive systematic review evaluating short- vs. long-course antibiotic therapy for streptococcal pharyngitis. Eight systematic reviews have been published to date [11,12,13,14,15,16,17,18]. These reviews include fewer clinical trials compared to our review, with a range of five to 22 studies. We used a broader search string to increase the sensitivity of the search and hand-searched reference lists of all these reviews to identify studies fulfilling our inclusion criteria. Another strength was the inclusion of papers in which confirmation of streptococcal infection was a requirement to be included in the study, then homogenizing the type of population these findings can be applied to.

The greatest weakness is the broad definition of inclusion criteria, hence including papers with high risk of bias, and grouping for meta-analysis papers comparing different types, length and doses of antibiotics. In the quality assessment, we found great weaknesses in the 50 studies. A total of 40 of the included studies had high risk of bias regarding blinding of participants (i.e., no blinding of patients, personnel and evaluators). These patients may be influenced by the investigator’s description of the trial. At the same time, the patients who agreed to participate could potentially have a more positive attitude towards a shorter and alternative treatment than the longer standard regimen. This could furthermore have influenced the patient’s own perception of subjective symptoms in the clinical evaluation and caused a systematic reporting bias. Problems in the blinding of the investigators could have influenced choice of treatment, and hence causing selection bias. Another major limitation is the high risk of publication bias.

It is problematic to group for meta-analysis studies comparing different doses. For example, five clinical trials compared short- vs. long-course penicillin V regimens, with daily doses ranging from 750 to 3200 mg. The pooled effect of short-term penicillin therapy on clinical cure showed a lower effectiveness compared with standard penicillin courses. For adults and teenagers, Gerber et al. [25] used 250 mg t.i.d. for 5 or 10 days; Strömberg et al. [26] 800 mg b.i.d. for 5 and 10 days; Zwart et al. [55,65] 500 mg t.i.d. for 3 or 7 days; and Skoog-Ståhlgren et al. [19] used 800 mg q.i.d. for five days in the experimental group and 1 g t.i.d. for 10 days in the control group. The use of such different doses of penicillin could explain the differences in the results observed in these studies as the effectiveness of β lactam antibiotics is dependent on time above the minimum inhibitory concentration [73]. We only included studies assessing short treatment regimens up to five days. Because of this, two studies, which compared amoxicillin six-day therapies to penicillin ten-day treatment, were excluded [74,75]. Finally, we only retrieved studies carried out in high income countries, making it more difficult to generalize to a worldwide setting, specifically those settings with high prevalence of rheumatic fever caused by GAS. Rheumatic fever remains a widespread disease all over the world, resulting in high morbidity and preventable early deaths in lower- and middle-income countries. Adequate and timely therapy could potentially prevent rheumatic heart disease [76].

### 5.3. Comparison with Previous Studies

Falagas et al. [18] assessed 11 trials comparing short- vs. long-course antibiotic treatment in patients with streptococcal pharyngitis/tonsillitis through a meta-analysis in 2008. They included studies comparing the same drug, in the same daily dosage, but for different durations. The review concluded that shorter courses of antibiotics, especially with penicillin V, had lower eradication rates than standard 10-day treatment. However, they did not find strong evidence for the best length of treatment. In response to this study, Dawson-Hahn et al. [16] suggested that we should shorten antibiotic treatment for common bacterial infections in outpatient settings. They also accentuated that shorter courses should be common practice in most other respiratory infections, such as pneumonia and acute bacterial sinusitis, but that currently, there was no clear evidence for shorter courses of penicillin for streptococcal pharyngitis. The meta-analysis by Altamimi et al. [15] from 2012 examined 20 trials comparing short- vs. long-course penicillin regimens in treating GAS pharyngitis in children. They found a comparable efficacy of the short duration (three to six days) treatment compared to penicillin V therapy for 10 days regarding clinical cure. Similar results were observed in a more recent review with the inclusion of 22 clinical trials [17].

The controversial results observed with all types of antibiotics were also observed with the cephalosporins. On one hand, in one meta-analysis published in 2007, the authors found that short-course cephalosporin treatment was as effective as long penicillin therapy regarding clinical cure; however, they raised the question whether such critical important antibiotics should be used on an infection as GAS pharyngotonsillitis as antimicrobial resistance is a growing concern [14]. On the other hand, in another review published two years earlier, cephalosporins were found to be more effective than long courses of penicillin to achieve early clinical cure, as we have also demonstrated in this review [12]. Another review published by the same authors found that short-term azithromycin courses were as effective as long-term penicillin treatment [13].

Finally, a Cochrane review published in 2016, which included trials based on type of antibiotics instead of length of treatment in patients with streptococcal pharyngitis concluded that penicillin should be the preferred first choice treatment for pharyngitis caused by GAS in both children and adults [77].

### 5.4. Relevance of the Study

The results of this review are highly relevant in clinical practice worldwide. For many years, the use of shorter courses of broad-spectrum antibiotics has been advocated as the right strategy to overcome the problems of treatment compliance without considering the harms regarding mild to moderate adverse events and the role of broad-spectrum antibiotics in the development of antimicrobial resistance.

Currently, the US Food and Drug Administration has approved cefdinir, cefpodoxime and azithromycin for a 5-day course of therapy for GAS pharyngitis. Macrolides and cephalosporins are considered as critically important antimicrobials for human medicine by the World Health Organization and should be reserved when the first-line choice fails [78]. In line with the recently released aware list [79], the first line option should be narrow-spectrum antibiotic such as penicillin V instead of broad-spectrum antibiotics such as azithromycin and 3^rd^ generation cephalosporines. Therefore, critically important antibiotics or antibiotics belonging to the watch group of the aware list should be used to a minimum in primary care in order to prevent development of antimicrobial resistance.

The findings of this review are highly relevant for funding future research addressing the optimization of the use of antibiotics in primary care. First of all, it should be discussed whether research including the use of critically important antibiotics should be funded and carried out in primary care. Not only could it be seen as unethical to use resources on comparing antibiotics that should not be used in primary care but also due to the higher risk of adverse events when exposed to the patients and development of antimicrobial resistance.

Finally, these findings showed the scarcity of available studies comparing only penicillin length and doses. Meta-analyses depend on the availability of high quality and homogeneous data to draw robust conclusions. Hence, future research should focus on executing trials that assess the effectiveness of different doses and lengths of penicillin across different contexts.

## 6. Conclusions

Macrolides and cephalosporins belong to the list of “Highest Priority Critically Important Antimicrobials”; hence, long-course penicillin V should remain as the first line antibiotic for the management of patients with streptococcal pharyngitis as far as the benefits of using these two types of antibiotics do not outweigh the harms of their unnecessary use.

## Figures and Tables

**Figure 1 antibiotics-09-00733-f001:**
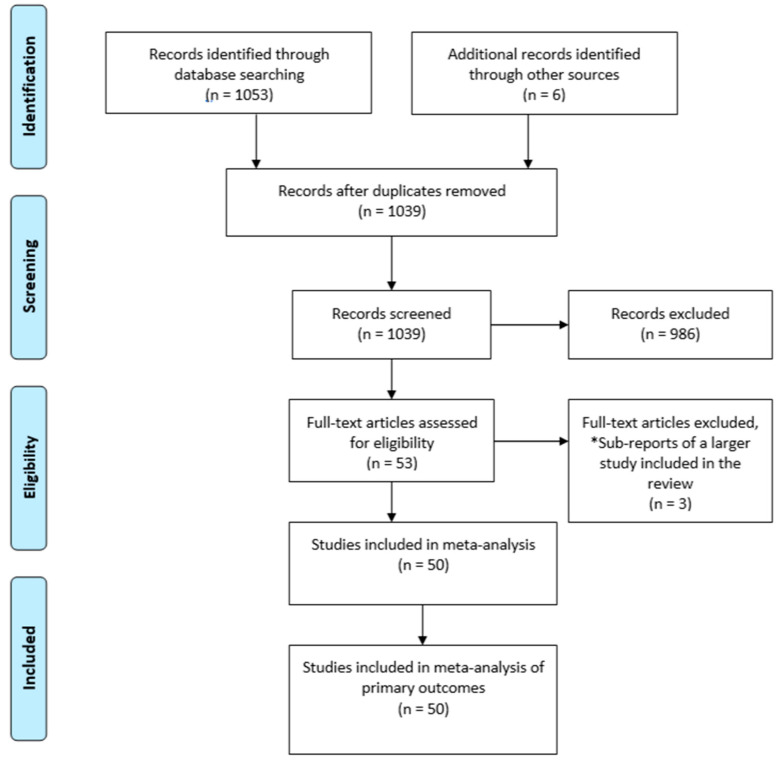
Preferred reporting items for systematic reviews and meta-analysis (PRISMA) flow diagram.

**Figure 2 antibiotics-09-00733-f002:**
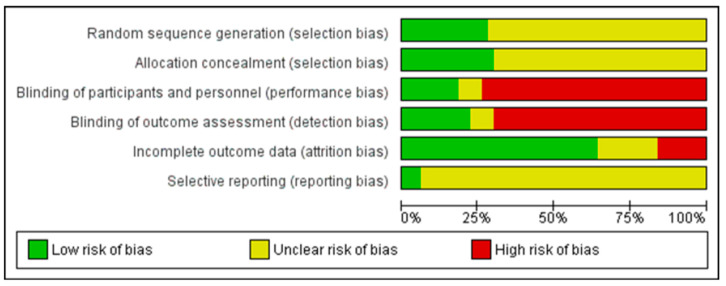
Bias assessment in the included studies. Risks are resented as percentages across different categories of bias. Unclear risk of bias (yellow) refers to studies with lack in systematic reporting of methods and results or when sources of funding were inadequately described. Green refers to low risk of bias; red refers to high risk of bias. Risk of bias: review authors’ judgements about each risk of bias item presented as percentages across all included studies. The main reason for unclear is lack of reporting, selective bias was only possible to assess in one trial with the protocol available in clinicaltrials.gov (Bethesda, MD, USA) and in other types of bias the source of funding for the project was unclear.

**Figure 3 antibiotics-09-00733-f003:**
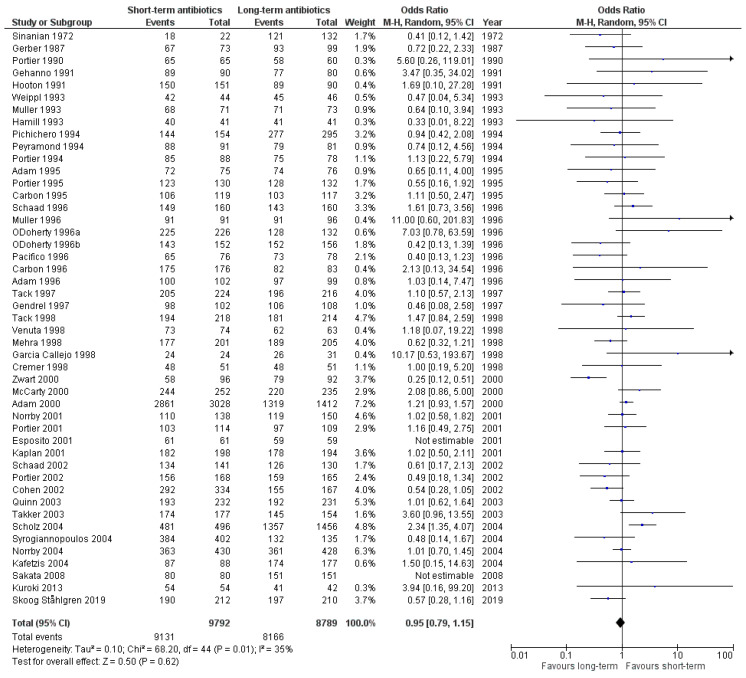
Early clinical cure of short-term antibiotic therapy compared to long-term regimen treatment.

**Table 1 antibiotics-09-00733-t001:** Characteristics of the 50 trials included in the meta-analyses.

Author, Year	No of Patients	Population: Age Range	Short-Term Antibiotic Therapy	Long-Term Antibiotic Therapy	Inclusion Based on Centor Criteria [72] (Yes/No)	Sponsorship
Sinanian, 1972	154	NA	Clindamycin 5d	Clindamycin 10d, Penicillin V 10d	No	The Upjohn Company, Michigan
Gerber, 1987	172	3–25	Penicillin V 5d	Penicillin V 10d	No	No mention of sponsor
Strömberg, 1988	203	7–70	Penicillin V 5d	Penicillin V 10d, Cefadroxil 10d	No	Bristol-Myers AB. Sweden and Leo AB Sweden
Portier, 1990	125	15–79	Cefpodoxime 5d	Penicillin V 10d	No	No mention of sponsor
Gehanno, 1991	170	5–70	Cefuroxime 4d	Penicillin V 10d	No	Glaxo
Hooton, 1991	241	NA (only included ≥16 year)	Azithromycin 4d	Penicillin V 10d	No	Pfizer Central Research Groton, Connecticut
Milatovic, 1991	209	NA (only included children)	Cefadroxil 5d	Penicillin V 10d	No	No mention of sponsor
Hamill, 1993	82	2–12	Azithromycin 3d	Penicillin V 10d	No	No mention of sponsor
Muller, 1993	144	NA (only included ≥12 year)	Azithromycin 3d	Clarithromycin 10d	No	No mention of sponsor
Weippl, 1993	90	2–12	Azithromycin 3d	Erythromycin 10d	No	No mention of sponsor
Peyramond, 1994	172	8–25	Cefixime 4d	Penicillin V 10d	No	No mention of sponsor
Pichichero, 1994	449	2–17	Cefpodoxime 5d	Cefpodoxime 10d, Penicillin V 10d	No	The Upjohn Company, Michigan
Portier, 1994	166	11–82	Cefpodoxime 5d	Penicillin V 10d	No	Roussel-Uclaf
Adam, 1995	151	1–12	Cefixime 5d	Penicillin V 10d	No	No mention of sponsor
Carbon, 1995	190	15–79	Cefotiam 5d	Penicillin V 10d	No	Roussel-Uclaf
Portier, 1995	262	8–30	Josamycin 5d	Penicillin V 10d	No	No mention of sponsor
Adam, 1996	201	3–17	Erythromycin 5d	Penicillin V 10d	No	Infectopharm Arzneimittel GmbH, Germany
Carbon, 1996	259	18–65	Azithromycin 3 and 5d	Roxithromycin 10d	No	Pfizer France
Muller, 1996	187	15–86	Azithromycin 3d	Roxithromycin 10d	No	No mention of sponsor
O’Doherty, 1996a	358	2–13	Azithromycin 3d (2 doses)	Penicillin V 10d	No	No mention of sponsor
O’Doherty, 1996b	308	NA (only included ≥12 year)	Azithromycin 3d	Cefaclor 10d	No	No mention of sponsor
Pacifico, 1996	154	3–12	Azithromycin 3d	Penicillin V 10d	No	Funded publicly
Schaad, 1996	320	1–14	Azithromycin 3d	Penicillin V 10d	No	No mention of sponsor
Gendrel, 1997	210	1–14	Spiramycin 5d	Penicillin V 7d	No	Rhone-Poulenc Rorer
Tack, 1997	440	1–18	Cefdinir 5d	Penicillin V 10d	No	Parke-Davis Pharmaceutical Research
Cremer, 1998	102	1–12	Azithromycin 3d	Cefaclor 10d	No	No mention of sponsor
Garcia Callejo, 1998	55	3–6	Azithromycin 3d	Amox/clav+Cefaclor 7-14d	No	No mention of sponsor
Mehra, 1998	396	3–13	Cefuroxime 5d	Cefuroxime 10d	No	No mention of sponsor
Tack, 1998	432	13–76	Cefdinir 5d	Penicillin V 10d	No	Parke-Davis pharmaceutical research
Venuta, 1998	137	4–12	Azithromycin 3d	Clarithromycin 10d	No	No mention of sponsor
Adam, 2000	4440	1–18	Ceftibuten 5d, Erythromycin 5d, Cefuroxime 5d, Clarithromycin 5d, Loracarbef 5d, Amox/clav 5d	Penicillin V 10d	No	Cascan, Essex Pharma, Glaxo Wellcome, Infectopharm Arzneimittel und Consilium, Lilly Deutschland, and Smith Kline Beecham Pharma
McCarty, 2000	487	1–12	Clarithromycin 5d	Penicillin V 10d	No	No mention of sponsor
Zwart, 2000	186	15–60	Penicillin V 3d	Penicillin V 7d	Yes	Funded publicly
Esposito, 2001	120	3–12	Cefaclor 5d	Cefaclor 10d	No	No mention of sponsor
Kaplan, 2001	392	12–61	Azithromycin 5d	Clarithromycin 10d	No	Abbott laboratories
Norrby, 2001	288	15–74	Telithromycin 5d	Penicillin V 10d	No	Aventis pharma
Portier, 2001	223	3–12	Josamycin 5d	Penicillin V 10d	No	Aventis Pharma
Cohen, 2002	501	2–12	Azithromycin 3d (2 doses)	Penicillin V 10d	No	Pfizer France
Portier, 2002	333	12–40	Clarithromycin 5d	Penicillin V 10d	No	Abbott France (Rungis, France) and Sanofi-Synthelabo
Schaad, 2002	271	2–12	Azithromycin 3d	Penicillin V 10d	No	Pfizer AG (PK)
Quinn, 2003	463	13–81	Telithromycin 5d	Clarithromycin 10d	No	Aventis Pharma
Takker, 2003	331	12–75	Clarithromycin 5d	Penicillin V 10d	No	Abbott Laboratories
Zwart, 2003	69	4–15	Penicillin V 3d	Penicillin V 7d	Yes	Funded publicly
Kafetzis, 2004	265	3–13	Cefprozil 5d	Penicillin V 10d, Clarithromycin 10d	No	No mention of sponsor
Norrby, 2004	858	NA (only included ≥13 year)	Telithromycin 5d	Penicillin V 10d, Clarithromycin 10d	No	Aventis Pharma
Scholz, 2004	1952	1–17	Cefuroxime 5d	Penicillin V 10d	No	No mention of sponsor
Syrogiannopoulos, 2004	537	2–16	Clarithromycin 5d (2 doses), Amox/clav 5d	Penicillin V 10d	No	Abbott Laboratories
Sakata, 2008	231	1–16	Cefcapene 5d	Cefcapene 10d, Amoxicillin 10d	No	No mention of sponsor
Kuroki, 2013	96	1–13	Amox/clav 3d	Amoxicillin 10d	No	Glaxo-SmithKline K.K.
Skoog Ståhlgren, 2019	422	3–67	Penicillin V 5d	Penicillin V 10d	Yes	Funded publicly

NA = Not available, d = days.

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
