# Peer review of "Short- vs. Long-Course Antibiotic Treatment for Acute Streptococcal Pharyngitis: Systematic Review and Meta-Analysis of Randomized Controlled Trials"

_antibiotics, 2020, doi:10.3390/antibiotics9110733_

Round 1

Reviewer 1 Report

The manuscript can be considered as an excellent review on the question if acute pharyngotonsillitis caused by Streptococcus pyogenes can be treated with short-course oral antibiotic regimens (≤5 days) compared to the established ≥7 days course of penicillin V.

Before publication, a few corrections / changes should be performed:

  1. Introduction, line 39: there, the taxonomically correct nomenclature for the colloquially used term “group A streptococcus” should be added, namely “Streptococcus pyogenes”. (Not all streptococci with the group A polysaccharide antigen in their cell wall are pyogenes, the common cause of streptococcal sore throat and the subject of this review!)
  2. Introduction, lines 40-41: Explain, what you mean with “... , specifically in high risk patients.”
  3. Data collection, lines 108-109: Add a reference or cite a website of the Cochrane ROB tool and version of this tool used!
  4. Line 114: add a reference to the website of WHO ICTRP!
  5. Line 115: correct to “(clinicaltrials.gov)”!
  6. Line 127: add a correct citation for the “Review Manager” program used and change text to “... which applies the Cochran-Mantel-Haenszel method ....”
  7. Figure 2: Check legend for correctness!
  8. Table 1: add a reference for the “Centor criteria” (6th collumn)!
  9. Discussion: Rheumatic fever (RF) is mentioned in only one sentence. An essential objective of the antibiotic treatment of streptococcal sore throat is the prevention of RF and rheumatic heart disease (RHD). This is also accepted as true for high-income countries and should be discussed a bit more detailed.

Author Response

See cover letter attached. Thank you.

Reviewer 2 Report

The manuscript compares the efficacy of long-term and short term antibiotic therapy in cases with acute pharyngitis caused by group A streptococcus.

The manuscript is well organized and clearly written. The criteria were well explained as well as possible bias of the included studies. It presents good overview of the literature published in the field of the discussed topic and also stressed that many of the available articles do not contain clear description of the material and methods.

In the introduction authors pointed that acute pharyngitis caused by group A streptococcus are self limiting in many cases. Could be nice if the authors recommend how to prolong the investigations in the field. 

The manuscript can be accepted for publication.   

Author Response

See cover letter attached. Thank you.

Reviewer 3 Report

This paper described antimicrobial treatment for acute streptococcal pharyngitis based on systemic review and meta-analysis.

First of all, I think this paper should be "research manuscript" not "review" section.

In the content, the research is well-organized and the manuscript is well-written. 

My concern was that the papers the authors analysed were old and only 2 papers after 2010 was included in this study. The authors should explain it.

Author Response

See cover letter attached. Thank you.

Reviewer 4 Report

It would be helpful to know if shorter v longer courses of treatment with the various regimens had any effect on transmission of streptococcal disease to family members or other close contacts, as this information could influence decision-making regarding which treatment to use.

Author Response

See cover letter attached. Thank you.

Reviewer 5 Report

The aim of the study entitled “Short- versus long-course antibiotic treatment for acute streptococcal pharyngitis: systemic review and meta-analysis of randomized controlled trials” was to determine how the type of antibiotics used and the duration of their use affect the therapeutic effect of acute streptococcal pharyngitis. As we are currently struggling with high resistance to antibiotics among microorganisms and great freedom in assigning antibiotics (e.g., non-adherence to recommendations), I consider the topic of the article current and important. Additionally, I would like to point out that the article is made very carefully.

I have only a few suggestions that I propose to introduce / consider:

- line 38: “However, the majority of these cases are viral and self-limiting even in cases caused by group A streptococcus (GAS)” -> The sentence may suggest mistakenly that GAS belongs to viruses. I would like to propose a minor modification, e.g. “However, the majority of these cases are viral and self-limiting even in cases caused bacteria belonging to group A streptococcus (GAS)”

- line 41: "high risk patients" -> please list a few examples in a bracket

- line 50: "Penicillin-resistant GAS has never been documented" -> please refer to the publication here

- line 73: "... languages other than spoken in America" -> please list them in a bracket

- line 77: "throat culture, rapid test or both" -> please specify what do you mean be a rapid test? (not all Readers may be familiar with the topic of GAS diagnosis)

Author Response

See cover letter attached. Thank you.
